# Contributions to a More Realistic Characterization of Corrosion Processes on Cut Edges of Coated Metals Using Scanning Microelectrochemical Techniques, Illustrated by the Case of ZnAlMg-Galvanized Steel with Different Coating Densities

**DOI:** 10.3390/ma17071679

**Published:** 2024-04-05

**Authors:** Marilia Fernandes Bolsanello, Andrea Abreu García, Luciana Xavier da Cruz Lima, Bruno Kneipel Neto, Jetson Lemos Ferreira, Jesualdo Luiz Rossi, Isolda Costa, Ricardo M. Souto, Javier Izquierdo

**Affiliations:** 1Instituto de Pesquisas Energéticas e Nucleares, IPEN/CNEN–SP, São Paulo 05508-000, SP, Brazil; mariliabolsanello@gmail.com (M.F.B.); jelrossi@ipen.br (J.L.R.); icosta@ipen.br (I.C.); 2Department of Chemistry, Universidad de La Laguna, 38200 La Laguna, Spain; andrea.abreu.28@ull.edu.es (A.A.G.); rsouto@ull.es (R.M.S.); 3ArcelorMittal Tubarão, Serra 29160-904, ES, Brazil; luciana.cruz@arcelormittal.com.br (L.X.d.C.L.); jetson.ferreira@arcelormittal.com.br (J.L.F.); 4ArcelorMittal Vega, São Francisco do Sul 89240-000, SC, Brazil; bruno.neto@arcelormittal.com.br; 5Institue of Materials and Nanotechnology, Universidad de La Laguna, 38200 La Laguna, Spain

**Keywords:** cut edge, corrosion, SVET, SECM, micropotentiometry, profilometry, SEM-EDS, ZnAlMg, steel, simulated acid rain

## Abstract

Corrosion processes at cut edges of galvanized steels proceed as highly localized electrochemical reactions between the exposed bulk steel matrix and the protective thin metallic coating of a more electrochemically active material. Scanning microelectrochemical techniques can thus provide the spatially resolved information needed to assess the corrosion initiation and propagation phenomena, yet most methods scan cut edge sections as embedded in insulating resin to achieve a flat surface for scanning purposes. In this work, the galvanized coatings on both sides of the material were concomitantly exposed to simulated acid rain while characterizing the cut edge response using SECM and SVET techniques, thereby maintaining the coupled effects through the exposure of the whole system as rather realistic operation conditions. The cut edges were shown to strongly promote oxygen consumption and subsequent alkalization to pH 10–11 over the iron, while diffusion phenomena eventually yielded the complete depletion of oxygen and pH neutralization of the nearby electrolyte. In addition, the cathodic activation of the exposed iron was intensified with a thinner coating despite the lower presence of sacrificial anode, and preferential sites of the attack in the corners revealed highly localized acidification below pH 4, which sustained hydrogen evolution at spots of the steel-coating interface.

## 1. Introduction

Enhancing the corrosion protection, safety and cost effectiveness of steel substrates is a global goal in building a more sustainable economy based on advanced materials. Only by protecting metals from degradation, thus extending the service life of steel components and improving energy efficiency, would the CO_2_ emissions be mitigated, associated with manufacturing, construction, and transportation of steel for the replacement of corroded structures, thus reducing the environmental footprint by minimizing rework and waste generation and reducing the impact of climate change [1]. In turn, corrosion aggravation requires energy-intensive restoration, increasing emissions of particulate matter, ground-level ozone, carbon monoxide, sulphur oxides, nitrogen oxides and lead. This self-perpetuating relationship highlights the need for effective metal coatings that break or slow down this recurring pattern.

After nearly two centuries of use, galvanized steels retain their indispensable role in the automotive, industrial and construction sectors by providing a surface finish that improves service life thanks to barrier and sacrificial protection mechanisms [2,3]. However, although alternatives have been proposed [4,5,6], no universal system has emerged that can replace zinc-coated steels in the mass production of vehicle bodies, due to their combination of corrosion resistance, ductility, and cost-effectiveness. At the same time, international automotive regulations demand improved fuel efficiency, which can be achieved by balancing the weight of the car body with the use of more resistant steel grades [7], without compromising safety standards. Reports show a 5.5% enhancement in fuel efficiency for a roughly 10% reduction in car weight [8]. Additionally, anthropogenic activity triggers the concentration of significant pollutants that can increase the aggressiveness of exposure conditions, particularly in densely populated urban and industrial areas with recurrent acid rain episodes [9,10]. Therefore, advances in steel galvanizing entail the development of more durable, reliable, and sustainable anticorrosive coatings, reducing the need for additional corrosion protection on coated components while also reducing costs and fuel consumption in automobiles.

Although protection against corrosion by zinc-rich coatings is mainly achieved by barrier effects and galvanic action, outdoor exposure progressively leads to the modification of the protection mechanism, which becomes controlled by the long-term deposition of passivating corrosion products, in particular zinc oxides, hydroxides and carbonates, which are strongly influenced by the chemical environment and pH [11,12]. In the search for more durable materials operating in increasingly aggressive environments, the addition of alloying elements to the molten bath, such as aluminium and magnesium, has proven to yield more efficient coatings for industry, particularly the automotive industry. These elements lead to the formation of different metallurgical products in the galvanized layer, as well as more compact and stable corrosion products [13,14]. Improved corrosion resistance compared with conventional hot dip galvanized steel is observed when Al is added in 5% (Galfan) or 55% (Galvalume) to the molten bath, and such enhancement has been overcome by the more recent incorporation of Mg [15,16,17,18]. The search for an optimized addition of both elements has yielded excellent anticorrosion performance for coatings containing Al and Mg at around 3% weight, which is appropriate for coatings applied in the automotive industry [19,20,21]. These elements also enhance the self-repairing effect in scratches and cut edges, as crucial regions under corrosive agents, because zinc ions are replaced by more protective Al and Mg corrosion products [20,22].

Frequently associated with the formation of stable layered double hydroxide (LDH) [23,24], the improved corrosion protection of ZnMgAl coatings, compared to pure zinc or binary zinc coatings, still presents unclear features. While salt spray and conventional electrochemistry of these alloys are frequently performed for their characterization [11,25,26], a comprehensive understanding of chemical interaction mechanisms requires the synthesis of micrometer- and nanometer-scale data from the activated sites [27], which is remarkable considering the microstructural heterogeneity of ternary-alloyed coatings [28].

Considering this scenario, the use of scanning probe microelectrochemical techniques [29] allows a more comprehensive evaluation of the performance of these advanced materials in harsh environments. The scanning vibrating electrode technique (SVET) makes it possible to measure local current densities in situ, with high spatial resolution, based on the detection of nanovolt potential differences [30] in a solution due to the ionic current flows generated by the electrochemical reactions that occur on a nearby electrochemically active surface. This method has been widely applied to study the distribution of cathodic and anodic currents that occur on the surface of galvanized materials, mainly at cut edges, in order to track changes over time [31,32,33,34] despite a very unfavourable surface ratio, down to 10^−6^, between exposed areas of protecting metallic coating and bulk steel. Next, the scanning electrochemical microscopy (SECM) technique has attracted interest in the characterization of local reactivity resulting from the exposure of vulnerable cut edges in the earlier stages of the corrosive attack [35]. The SECM technique includes a rastering microelectrode, capable of either amperometrically detecting redox species that will be converted at the probe, such as molecular oxygen dissolved in an aqueous solution, or potentiometrically detecting pH gradients developed as a result of the corrosion reactions. Although the chemical characterization achieved with the SECM and micropotentiometry has greatly advanced the understanding of cut edge systems and their reactivity [32,35,36,37,38,39], most studies using SECM and SVET have proceeded by embedding the cut edges in insulating resin to provide a fairly controlled surface configuration although producing an unrealistic metal ratio exposure.

In this work, a more realistic approach was tested to evaluate the effect of simulated acid rain on cut edge systems, exposing not only the cut edge but also the galvanized sides to the aggressive environment, thus attaining a coating/substrate ratio that better resembled the actual exposing conditions at artificially produced cut edges. The strategy proceeds as an adaptation of an arrangement previously developed in literature for SVET analysis [40,41,42]. The SECM operated in redox competition and potentiometric modes, allowing dissolved oxygen and pH distributions to be correlated with local reactivity. Additionally, the macroscopic behaviour of the system was characterized using conventional electrochemical techniques and surface analysis, namely Scanning electron microscopy (SEM), Energy dispersive X-ray analysis (EDS) and Glow-discharge optical emission spectroscopy (GDOES) as ancillary methods to monitor microstructures and identify elements and compounds in the ZnAlMg coating layer. This combination of methods links microelectrochemical corrosion assessment to conventional coating performance assessment, ensuring that it meets corrosion protection standards and remains effective under harsh conditions. Also, to seek corrosion resistance with optimized efficiency, three different coating densities of the galvanized layer were tested, revealing that performance is determined by factors that extend beyond coating thickness.

## 2. Materials and Methods

### 2.1. Materials and Reagents

The tests were carried out on double-sided ZnAlMg galvanized steels, commercially known as Zagnelis^®^ Surface, supplied by ArcelorMittal (Tubarão, Serra, ES, Brazil) [43]. The substrate consisted of DX57D (European Steel and Alloys Grades number 1.0853; BS EN 10346:2015) [44], an interstitial free ferritic, low-carbon and low-nitrogen steel with 0.72 mm thickness well-suited for cold forming and deep drawing applications. The composition of the steel sample, according to the manufacturer, is given in Table 1. The steel was coated by continuous hot-dip galvanizing in a molten bath consisting of 93.5% of zinc, 3.0% of magnesium and 3.5% of aluminium. The surface finish is transferred during the skin pass or temper rolling section. Coating densities of 70 g m^−2^ (named ZM70), 90 g m^−2^ (ZM90) and 120 g m^−2^ (ZM120) were applied on both sides, resulting in estimated coating thicknesses of 5, 7 and 10 µm, respectively, according to the expected coating density and assuming a homogeneous distribution. The materials were received in the form of 500 mm × 500 mm steel sheets. The samples for SEM analysis were cleaned using a 5% weight glycine solution in deionized water for oxide removal and inhibitor protection [45].

Simulated acid rain (SAR) was used as a test electrolyte to reproduce aggressive conditions, comprising 0.2 g L^−1^ NaNO_3_, 0.2 g L^−1^ Na_2_SO_4_, and 0.2 g L^−1^ NaHCO_3_, adjusted to pH 4 using minute amounts of H_2_SO_4_ and NaOH [46]. To evaluate the anticorrosion properties and surface morphology of the galvanized coating, 20 mm × 20 mm coupons of ZM70, ZM90 and ZM120 were precision cut using a linear saw Isomet-1000 (Instrumental Brasil, São Paulo, Brazil), to minimize deformation. Then, to mitigate edge effects, the cut edge samples were ground using an MD-Gekko metallographic grinding machine (Struers, Champigny-sur-Marne, France) with P320, P500, P800, and P2000 grit sandpaper, followed by metallographic polishing using a cloth with 6 and 1 µm diamond paste (Arotec, Cotia, Brazil).

### 2.2. Surface Characterization

The macroscopic surface morphology was first studied using a Profilm 3D-filmetrics optical profilometer from Scientec Ibérica (Madrid, Spain), based on interferometry with a lateral resolution of 50 nm. The roughness estimates were averaged for at least 9 measurements in random areas of dimension 0.9 × 0.9 mm^2^. The surface microstructure and composition were then analysed by scanning electron microscopy combined with Energy dispersive X-ray spectrometry (SEM/EDS) based on a JEOL JSM-7100F thermal field emission electron microscope (Japan Electron Optics Laboratory, Tokyo, Japan). The objective lens of the JSM-7100F produces a maximum probe current of 200 nA and does not form a magnetic field around a sample, allowing visualization of magnetic ferritic steels with a resolution of 1.2 nm. The images were acquired using backscattered electrons. The equipment was operated using AZtec software V. 5.1, supplied by Oxford Instruments (Abingdon, UK). Additionally, EDS semiquantitative chemical composition maps were obtained using the same equipment with a NordlysMax^2^ detector supplied by Oxford Instruments. SEM/EDS analysis was performed on cross-sectional areas using 30 mm × 5 mm rectangles embedded in thermosetting phenolic hot resin with a Struers CitoPress-10 mounting press. To improve electrical conductivity and minimize sample charging effects during analysis, the embedding blocks were coated with a thin layer of metallic gold, ranging from 2 to 5 nm, using a sputtering deposition system (IB-19500CP Cross Section Polisher, JEOL).

Analysis of the in-depth chemical composition of the coated surface of samples ZM70, ZM90 and ZM120 was carried out using a Pulsed radio frequency GDOES spectrometer with Differential interferometry profiling (GD Pofiler 2, Horiba Scientific, Kyoto, Japan). The samples were placed against the O-ring facing the anode tube in an argon atmosphere, without requiring ultra-high vacuum conditions. The equipment used a 13.56 MHz pulsed RF plasma source, with optimized pressure control during the sputtering process using two pumps in the plasma source.

### 2.3. Conventional Electrochemical Characterization

Conventional electrochemical measurements were carried out using a potentiostat/galvanostat model 600+ (Gamry Instruments, Warminster, PA, USA), controlled by a personal computer. To ensure test reproducibility and avoid edge effects, an area of 10 mm × 10 mm was isolated using beeswax. The electrochemical response of the coating was analyzed by first allowing the open circuit potential (OCP) of the samples exposed to the SAR solution to stabilize over a 3000 s immersion period. Subsequently, polarization measurements were performed in triplicate for each sample type and polarization direction at a sweep rate of 1 mV s^−1^. The anodic polarization curves were started from OCP − 20 mV up to OCP + 1.0 V, while the cathodic polarization curves were initiated from OCP + 20 mV down to OCP − 1.0 V. A Saturated calomel electrode (SCE) and a platinum auxiliary electrode served as the reference and counter-electrode, respectively. All potential data reported here refer to the SCE. Graphical interpolation was employed to determine the corrosion current and Tafel slopes for the anodic and cathodic branches.

Electrochemical and surface analysis of the cut edge system exposed in the SAR solution was performed using 30 mm × 5 mm rectangular coupons, cut using a diamond blade saw. OCP measurements were first recorded vs. SCE using these samples by immersing the edge until an approximate depth of 1 or 10 mm in the SAR solution. The samples were held vertically, avoiding contact of the electrical connection with the electrolyte.

### 2.4. Microelectrochemical Characterization

For microelectrochemical analysis, samples were mounted in a dedicated polymer sample holder with a 1 mm × 5 mm opening at the bottom centre, so that the rectangular sample could be inserted and secured using fast-curing silicon paste and remaining vertically allocated for scanning the top side (i.e., the horizontally exposed cut edge). An approximate depth of 10 mm of the sample length was introduced and sealed in the holder opening, thus avoiding further contact with the electrolyte and fixing its position. To optimize the coating-substrate area ratio, only the top cross-section of the cut edge was exposed, and the remaining edges (placed vertically) were meticulously sealed with silicon paste; therefore, only the galvanized surfaces at the vertically placed sides, and the cut edge at the top (placed horizontally) were exposed. Thus, a cut edge iron-based surface ca. 3.6 mm^2^ was concurrently exposed in the cell along with the coated sides, accounting for almost 200 mm^2^ (iron/coating ratio over 50). The target holder was completed with a cylindrical methacrylate wall of 3 cm inner diameter and 4 cm height, sealed with silicon paste, resulting in an electrolyte container for approximately 25 mL of solution. This sample holder was used for both SECM and SVET measurements.

SVET equipment was supplied by Applicable Electronics (New Haven, CT, USA). The sensing probe utilized was a needle 10 µm diameter and 2 cm long composed of 80% platinum and 20% iridium. The microprobe body was insulated with paralene C^®^, while the uninsulated tip was platinized by electroreduction in a hexachloroplatinic acid solution, resulting in a spherical platinum black deposit that nearly doubled the electroactive area. To minimize noise interference, the bidirectional tip vibration was tuned to bimorph frequency, approximately 185 Hz in the *X* direction and 78 Hz in the *Z* direction. The vibrating microprobe registers potential gradients between two locations within the electric field, and then it is transformed into a sinusoidal AC output signal with the same frequency as the vibration, so that a lock-in amplifier can convert the amplitude of the signal into ionic current densities. This allows the detection of local anodic or cathodic activation sites on the corroding surface [30]. Measurements were conducted in the *XY* plane at a constant distance of 150 µm in the *Z* axis from the cut edge surface under investigation, all monitored using a video zoom camera placed vertically over the sample.

The same sample arrangement was employed for SECM measurements using an instrument built by Sensolytics (Bochum, Germany). For amperometric measurements, 10 µm diameter platinum disk microelectrodes embedded in a glass capillary were used. The cell was completed with a Pt wire as counterelectrode and an Ag/AgCl/KCl(sat.) reference electrode. The amperometric measurement mode was the redox competition operation, with the Pt probe polarized at −0.75 V vs. SCE to monitor the consumption of dissolved molecular oxygen due to the cathodic process [47], while scanning at an approximate distance of 50 µm from the sample. A micro-camera was placed in a horizontal position to aid tip positioning at the desired height, taking advantage of the transparent methacrylate cell walls.

SECM potentiometric measurements were performed using homemade antimony microelectrodes with an active disk of approximately 20 µm in diameter. Sb/Sb_2_O_3_ electrodes are known to be pH sensitive electrodes and their fabrication was carried out as previously described in the literature [35]. Briefly, borosilicate capillaries (wall thickness 0.225 mm, outer diameter 1.5 mm) supplied by Hilgenberg GmbH (Masfeld, Germany) were pulled using a model P-30 micropipette puller (Sutter Instrument, Novato, CA, USA). Then, Sb fibre (2.0 cm in length and approximately 20 μm diameter) was inserted into the lumen of the micropipette with the tip protruding approximately 5 mm, and Cu wire (approximately 12 cm in length and 0.5 mm in diameter) was inserted at the other end to provide an electrical contact that was ensured with a small quantity of liquid mercury metal. Both ends of the micropipette were sealed with Loctite^®^ adhesive. A linear relationship was obtained between solution pH and potential, with a slope of −44 mV per pH unit, in the range 3 ≤ pH ≤ 11. Although the slope of the calibration line is smaller than the Nernstian slope, this is not unusual for these single-barrel microelectrodes [35,47]. The Sb/Sb_2_O_3_ probe was then used in combination with an Ag/AgCl/KCl(sat.) reference electrode for SECM potentiometric tests. The Sb/Sb_2_O_3_ probe was placed at a height of approximately 50 μm above the sample as adjusted using the video-camera system.

## 3. Results and Discussion

### 3.1. Surface Characterization

The as-received samples presented a heterogeneous appearance at the millimetre range, as confirmed by optical profilometry. Figure 1 shows illustrative images of the samples, which exhibit flake-like morphologies similar to features reported in the literature for ternary galvanic coatings [48,49,50]. The black spots seen in the images result from the denoise algorithm used for plotting and do not correspond to actual features appearing at the surface. Interestingly, the most compact coating was found to be ZM70, and the least homogeneous coating was found for ZM90. Roughness parameters also reported a higher Sq value for ZM90 (4.21 ± 0.25 µm, 20 measurements) compared to ZM70 (3.66 ± 0.25 µm, 9 measurements) and ZM120 (3.34 ± 0.26 µm, 15 measurements). The deepest flake-like feature found in the ZM70 system (see the lower left side of the image) showed approximately 6 µm depth, with the remaining features having a step height of approximately 4 µm. This is comparable to the expected thickness of the galvanized coating, estimated at 5 µm for this lower galvanic content. However, the samples were completely covered, as demonstrated by optical microscopy observations and their electrochemical behaviour, suggesting that the coating had a thicker dimension than theoretically estimated. Furthermore, various microstructural defects and grain accumulation must account for such heterogeneous distribution, and the coating density values may differ from what was expected depending on the coating composition.

The microstructure of the ZM70 galvanized coatings revealing distinct morphology is also supported by the FEG-SEM images given in Figure 2a,b, obtained after the glycine cleaning procedure. Figure 2c shows the semiquantitative elemental EDS map, as well as the individual elemental maps. The position where SEM image in Figure 2b was recorded is also labelled with white squares in the EDS maps of Figure 2c. Despite the cleaning process, white corrosion products (labelled 1 in the figure), with spongy and porous morphology, can be observed clustered at different points at the top of the image in Figure 2a, resulting from the atmospheric exposure. Small, slightly rounded craters (label 2), measuring between 4 and 6 µm, were observed, possibly due to the formation of zinc oxide clusters that were subsequently removed during prior cleaning of the material. The black rust seen in label 3 is a precursor to the tarnishing process. Zinc particles between 7 and 10 µm are randomly dispersed on the surface (label 4), while zinc dendrites (label 5) and eutectic phases (label 6) are also observed [49,50]. The elemental distribution of oxygen on the surface accompanies the presence of Al and Mg as more reactive elements, thus leaving areas of metallic zinc readily exposed to the environment. 

Interesting observations can be made by cross-referencing the SEM image seen in Figure 2b, and the location of the corresponding area in the EDS maps of Figure 2c (white squares) with the aluminium and magnesium contents revealed by EDS in Figure 2c. At the spot labelled 7, a distinct Zn-MgZn_2_ binary region is evident, characterized by zinc flakes intercalated with magnesium-rich phases. The presence of white, dense, porous oxide above the zinc layers indicates that this specific region has undergone atmospheric corrosion, resulting in the formation of zinc oxides. Additionally, there is a noticeable reduction in Mg-rich beds, which appear thinner and narrower. This observation proves that the darker areas of the intermetallic MgZn_2_ were selectively corroded. The dissolution process likely occurs through a localized corrosion mechanism, where the zinc-rich phase acts as a site for the cathodic oxygen reduction reaction, while the anodic reaction involves the dissolution of the MgZn_2_ phase. This suggests that specific regions within the galvanized coating have a different composition and are therefore likely to react differently in a corrosive environment.

Another interesting aspect to consider is the distinct behaviour of aluminium precipitates within the galvanized coating (label 8). These Al-rich intermetallics are found in ternary eutectic regions without defined morphology, often interspersed with irregular lamellae of zinc- and magnesium-rich phases. The presence of a large, elongated aluminium deposit is observed adjacent to the zinc dendrites. This particular pattern of aluminium distribution suggests the formation of complex microstructures, probably due to variations in the cooling rate during the galvanizing process. The presence and arrangement of these aluminium-rich regions may have implications for the overall corrosion resistance and mechanical properties of the galvanized steel.

The surfaces obtained with a denser galvanic coating exhibited similar features as those observed in Figure 3 and Figure 4. Zinc dendrites flanked by binary and ternary eutectics were observed in all samples. Significant accumulation of oxides is observed near the aluminium dendrites in the ZM90 samples, which could further compromise the protective properties of the galvanized coating, as they originate regions more vulnerable to corrosion attack [14]. Additionally, ZM90 and ZM120 also exhibited aluminium precipitation in the form of elongated particles up to 5 µm in length, in a narrow region between zinc dendrites (seen in Figure 2 and Figure 3). This precipitation can be caused by element segregation during the solidification process of the coating, which depends on the cooling rate as well as the composition and galvanizing conditions [51].

The cross-section of the ZM70 sample was analysed using FEG-SEM and EDS, as depicted in Figure 5. The SEM images in Figure 5a,b clearly reveal the coating layer in brighter colour than that of the steel substrate (seen in label 1), with an average thickness of approximately 8–10 µm. This layer is significantly greater than the previously mentioned estimate, resulting from the lower density due to a porous structure that has flaws and fissures on its surfaces, characteristic of the fragile, dark oxides that form in the galvanic layer during atmospheric exposure. This finds a correlation with the heterogeneous profilometry data given in Figure 1a for this system.

As revealed by the EDS images (see Figure 5c), Zn is the main element of the coating (label 2), while Al and particularly Mg are mainly present in the zinc dendrites, easily identifiable as ellipsoidal shapes (label 3); they are surrounded by a darker coloured lamellar phase. Zn is present in all of these structures, while oxygen is almost absent, as revealed by the EDS map of elemental oxygen, evidencing the presence of intermetallic alloyed components. Additionally, the oxygen map reveals a nearly continuous oxide layer along the iron-coating interface, while highlighting the absence of significant oxide formation in deeper regions of the coating.

The coarse, elongated lamellar morphology seen in the SEM images as spots (4) in Figure 5a,b represents binary eutectics, where bright zinc lamellae are interspersed with MgZn_2_, displaying an intermediate grey colour. This is supported by the richer Mg content presented in the Mg elemental map in Figure 5c. On the other hand, the most irregular phase with darker lamellae corresponds to the Zn-Al-MgZn_2_ ternary eutectic (label 5) which is richer in aluminium and reveals a coarse pattern [11,14,51,52]. Interestingly, this aluminium-containing ternary phase appears deeper, around 1.6 to 1.8 µm from the surface of the coating, and then extends to the interface of the galvanized coating with steel, precisely at the interface between the different lamellar orientations, marked with red arrows [11,14,27]. Additionally, the small dots in spot 6 are found to correlate with zinc oxides resulting from atmospheric exposure.

Similar structures were obtained with the ZM90 samples, with an apparently limited presence of ternary eutectic as seen in Figure 6. Mg was particularly observed at the iron-coating interface and in binary eutectic regions. The occurrence of discontinuities appeared more heterogeneously distributed in the cut edges of ZM120 shown in Figure 7, where oxides were observed along the deeper parts of the coating. Considering that greater homogeneity and compactness would result in less microgalvanic interactions between pure Zn and intermetallic phases, although the mechanical properties of the coating would benefit from the precipitation of dendrites and secondary phases, the ZM70 sample apparently better satisfies a compromise situation.

Surface composition was quantified using GDOES analysis to determine global trends in element accumulation near the surface and underlying substrate, and the results are shown in Figure 8 to a total depth of 20 µm. The graphs on the left side of each row show a data overview, and subsequent plots show magnifications for closer inspection of the elemental trends in the outermost layer (see the plots in the centre) or for the elements present in lower contents (right side).

In general, the Zn content increases sharply over the first 500 nm as the oxygen content decreases, exceeding 95 at.% Zn at 1 µm depth in all samples. This dominant Zn content begins to decrease around 5.1 µm for ZM70 (Figure 8a, left), which is the criterion conventionally adopted for estimating the average coating thickness. This value is similar to that theoretically estimated during sample preparation, although the GDOES measurement focuses on a single point and should, therefore, be considered with care. The presence of iron ends up exceeding the zinc content at ca. 9.4 µm in the ZM70 system (Figure 8a, left), this depth being representative of efficient galvanic protection. Concerning the ZM90 coating (Figure 8b), the chemical composition does not significantly differ from the thinner coating, and the maximum Zn content, as well as the higher presence of iron, are observed around similar depths of 5.1 and 10 µm. However, these values turned out to be poorly reproducible when considering GDOES measurements taken at other spots, which may indicate heterogeneous deposition and therefore lead the occurrence of asymmetries detrimental for anticorrosion properties. Finally, the ZM120 system matches the expected thickness ranges, as iron was not detected until the depth reached approximately 8.4 µm and did not become the major component until 13.7 µm (Figure 8c).

The expanded analysis presented in the middle plots of Figure 8 reveal that zinc coexists with oxygen in all samples within the first quarter micrometre, while only traces of aluminium and magnesium are observed. Zn hydroxides may well become the major component of ZM90 and ZM120 in the first 100 nm, as suggested by the stoichiometric ratio, of almost 2:1 in favour of Zn in central graphs of Figure 8b,c. In contrast, the Zn surface exhibits a greater metallic character in the ZM70 coating due to the lower oxygen content, which reaches almost 15%. The percentage of oxygen in the first 0.4 µm is actually double when considering the ZM90 and ZM120 coatings, probably due to the enlargement of the coating and the consequent increase in the tendency for oxide formation.

The lower density of pure aluminium (2.38 g cm^−3^) and magnesium (1.57 g cm^−3^) compared to zinc (6.92 g cm^−3^) should allow greater migration for these two elements during the deposition of the molten metal during galvanizing. However, the influence of density is not the only factor acting on these metals since considerable amounts of aluminium and magnesium are also observed near the Fe-Zn intersection in the right graphs of Figure 8. This observation correlates well with the inspection of the cut edges using the SEM-EDS images of Figure 5, Figure 6 and Figure 7, which show the accumulation of Al and mainly Mg at the iron-coating interface. This phenomenon may be related to the fact that, during the hot dip coating process of galvanized steel with the molten bath, aluminium and magnesium tend to mainly form intermetallic compounds precisely at the steel-coating interface. Consequently, a wide range of intermetallics can be developed, such as Zn-Fe, Al-Fe and Mg-Fe interlayers, as well as those involving more than one element of the coating, such as Fe_2_Mg_5_Zn_2_ (σ-phase) and FeMgAl_6_ (β-phase). In addition, the significant amount of oxygen observed at this interface can be associated with the oxidation of the steel in a stage prior to coating. Even when cleaned before hot dipping, DX57D steel reacts quickly with air to form iron oxides. At the nanometre scale, as occurs in the GDOES analysis, a few minutes between the cleaning stage and the immersion in the molten zinc are sufficient for the formation of the first iron oxides.

### 3.2. Electrochemical Behaviour

The coated samples were first tested by performing dynamic polarization to determine the anticorrosion effectiveness of the galvanized steel within the first hour of immersion. The polarization curves are shown in Figure 9a, revealing a corrosion potential below −0.90 V vs. SCE, typical of Zn-based surfaces. The vulnerability to corrosive attack is evidenced by current values exceeding the mA cm^−2^ range under anodic polarization. Table 2 shows the corrosion parameters as interpolated graphically from the Tafel analysis of data in Figure 9a. The corrosion current densities reveal good performance around 3 µA cm^−2^, which translates into a degradation rate of ca. 1 × 10^−5^ mm per year, which is about 2–3 magnitudes lower than expected for mild steel under acidic rain conditions [53,54]. The highest efficiency was achieved with the denser ZM120 system, although ZM70 appears to be more stable than the ZM90 system, as suggested by the lower corrosion current density.

Our main interest in the investigation of this type of system is focused on achieving efficient protection and longer life of the cut edge under realistic immersion conditions before the initiation of the corrosive attack. Thus, the cut edge of the 30 mm × 5 mm coupons of each sample was exposed vertically to the SAR test solution at a depth of approximately 1 and 10 mm, and the open circuit potential (OCP) with respect to the SCE was recorded. Although this measurement configuration does not reflect a conventional electrochemical analysis, the observations and recorded data revealing the surface state and spontaneous evolution were interestingly dependent on the anode-to-cathode surface ratio exposed in SAR solution. This manner, Figure 9b shows the results of the ZM70 at the two immersion depths mentioned above, in comparison with the ZM90 system immersed at the biggest depth (i.e., 10 mm). When the ZM70 sample established minimal contact (1 mm depth, black line in Figure 9b) with the SAR solution, the result revealed a transient potential with a maximum (nobler) OCP value, attained between 315 and 375 s of exposure. The OCP then decreases again, revealing a greater thermodynamic tendency to oxidation until 20 min, before the potential eventually returns to less negative values. This observation suggests the initial formation of oxides in contact with the SAR solution, and it is likely that once these oxides cover the iron surface at the cut edge, the potential will be closer to that of reactive zinc. Conversely, when the sample is more deeply immersed in the solution, the potential evolution responds more smoothly, regardless of the thickness of the coating (see the red and blue lines in Figure 9b corresponding to ZM70 and ZM90). This suggests a progressive formation of zinc (hydro)oxides on the galvanized faces in full contact with the solution, under galvanic interaction with the iron-based cut edge, meeting the expected protection method.

To locally characterize the phenomena that actually occur at the cut edge of the galvanized coating system, the surfaces of the cut edges were studied using microelectrochemical methods. SVET analysis was used to visualise the localized reactivity and ionic current densities reflecting anodic and cathodic activation at the cut edge, as the technique allows the detection and differentiation of anodic and cathodic ionic current densities, as well as the acquisition of optical images using the camera placed above the sample. The measurement procedure resulted in the images in Figure 10 for the ZM70 system. Measurements were made with the probe travelling in the *XY* plane over the cut edge at a constant distance of 150 µm in the *Z* direction, revealing cathodic activity on the iron-based surface. Since iron is a more noble metal than Zn, Al and Mg, it is expected to sustain the cathodic half-reaction as part of the degradation process both in acidic or neutral and alkaline conditions (see Equation (1)). This causes the consumption of hydrogen ions or the release of OH^−^ ions (as a function of pH), which renders either the consumption of positive charges (H^+^) or the production of negatively charged species (i.e., OH^−^), responsible for the negative current densities observed along the *Z* axis.
O_2_ + 4 H^+^ + 4 e^−^ → 2 H_2_O or O_2_ + 2 H_2_O + 4 e^−^ → 4 OH^−^(1)

The evolution of the ZM70 system over time reveals that cathodic activity starts quickly and intensifies throughout the initial 24 h (notice that all scans are plotted using the same scale) before the progressive decay of the activity. The reason for the subsequent decay in reactivity relates to the development of protective passive film of corrosion products covering the reactive surface. Indeed, the optical inspection reveals the precipitation of white corrosion products on the exposed iron, thereby hindering surface reactivity. Care should be taken when quantitatively evaluating the SVET results discussed here, as the high ionic strength of the SAR solution prevents precise estimation of ionic current densities, with the electrolyte buffering the developed gradients of ionic charges (which results in ionic current density) in the vicinity of active sites. Therefore, although the SVET measurements with the ZM90 and ZM120 coatings were carried out, collecting very similar observations, their significance and comparison are quite limited and will not be discussed in detail in this work.

Over time, the observed cathodic activity becomes quite localized to specific locations where apparently no oxide deposits form (see magnification in Figure 10d); however, all exposed steel continues to behave cathodically. Based on the chemical nature of the galvanic corrosion process, it is expected that the active metals dissolve from the galvanized sides composing the coating, and that their diffusion onto the cathodically activated steel surface results in inhomogeneous precipitation of hydroxides. These precipitates have a partially protective and insulating character, nonhomogeneous coverage would lead to greater cathodic reactivity in places where there are no corrosion products covering the steel surface. The mechanism involves diffusion and subsequent encounter of metal cations (mainly Zn^2+^, Mg^2+^ and even Al^3+^) with a sufficient OH^−^ concentration from half-reaction (1), resulting in the precipitation of the metal hydroxides. The advancement of the front of white (i.e., Zn-based) products over the cut edge has already been reported in the literature when the cut edge was embedded in an insulating resin [32,33,35,36,37,38,39]. The local cathodic activation observed at specific surface spots, rather than a front advance, seems to occur as a result of the more realistic exposure conditions herein considered, with the system also exposing the galvanized sides to the aggressive environment.

A weak anodic activity (revealed by positive ionic current densities) is observed near the edges using SVET. This is more clearly observed in Figure 10c, with current densities above 0, always originating from the coating sides and not from the iron surface at the cut edge. Additionally, the rapid diffusion of hydrogen ions can make it difficult to actually monitor the concentration gradients responsible for the ionic current signal, which is further hampered by the high conductivity of the SAR solution. Another interesting feature was revealed by the optical micrographs taken in situ immediately after the SVET measurements, as bubbles randomly formed at the iron-coating interface when exposed to the SAR solution, which must account for the evolution of hydrogen gas resulting from a complementary cathodic process:2 H^+^ + 2 e^−^ → H_2_(2)

The effect on SVET recording is equivalent to the main cathodic reaction (1) that occurs on iron, since this technique is not capable of discerning the chemical nature of the species involved as long as their diffusion favours ionic current densities flowing in the same direction. For this generation of hydrogen to occur, it is necessary to attain a sufficiently acidic environment, which could only become achievable locally after the active release of metal cations, to a sufficient degree so that their hydrolysis can lead locally to significant acidification [55]. In fact, as the coating at edges can become thinner and even promote asymmetries due to different grain structures, these sites are more vulnerable to corrosive agents, leading to severe localized corrosion [56]. Although any of the metal cations that dissolve from the galvanized coating can undergo hydrolysis, given the strongly acidic nature of the Al^3+^ cation and its accumulation at the iron-coating interface (along with Mg), this species is considered to be the most likely responsible for the local acidification:Al^3+^ + 3 H_2_O → Al(OH)_3_ + 3 H^+^(3)

Additional analyses are needed to more precisely discern the nature and origins of such an acidification resulting in hydrogen evolution, since a microelectrochemical characterization under hydrogen gas evolution is subject to disruptive convective effects. This convection not only prevents proper SVET analysis, but SECM in generation–collection mode also becomes limited, although it has been reported to be qualitatively applicable as an approach for the determination of hydrogen evolution [29,56].

To discern the chemical nature of the species involved in the surface reactions at the cut edge, the SECM technique was employed for electrochemical visualisation of the environment in the electrolyte solution adjacent to the metal surface. The use of the sample generation and tip collection mode (SG-TC SECM) for hydrogen gas detection did not provide clear results in the SAR solution due to the intense convection caused by the movement of the hydrogen bubbles. However, the redox competition mode (RC-SECM) served to determine the surface reactivity towards oxygen consumption according to reaction (1). The results in terms of minus tip current (i.e., absolute cathodic current in nA) are shown in Figure 11 for the ZM70 system as the exposure time in the SAR solution progresses. This operation mode allows obtaining qualitative images of the cathodic activity on the iron exposed at the cut edge, since iron, being a nobler metal than the elements which make up the galvanic coating, would sustain the cathodic reaction of the degradation process.

The RC-SECM maps show that oxygen is competitively consumed near the iron surface, almost entirely in the middle, causing the tip current to drop to almost zero. This is readily observed at the start of the experimental series (see Figure 11a), accounting for a rapid cathodic activation of the surface, which appears homogeneous over the exposed iron. The tip current recorded away from the surface at the same height (i.e., *Z* value) should actually correspond to the limiting cathodic current reached in the bulk solution in the presence of dissolved molecular oxygen, because there is no surface under the probe when it is moved laterally away from the central iron-based region, which is different to the conventional SECM arrangement employed in the literature until now. Next, as time passes, the oxygen consumption rate apparently decreases after 3 h of immersion, according to the nonzero current values attained at the centre of the scan in Figure 11b. This is more clearly revealed in Figure 12a, where the evolution of the middle line of the maps in Figure 11 (at *X* = 500 µm) are plotted in terms of normalized current values.

The dynamic evolution of the cut edge system is revealed by the variations in the current measured at the top centre of the cut edge. It is likely that within the first hours of immersion, corrosion products are formed from the dissolution of the galvanic coating, which are next deposited on the surface of iron, partially hindering its cathodic reactivity. Furthermore, this effect is asymmetrical, since the cathodic currents reached at each side of the coatings exhibit major differences, as seen in Figure 11 and more clearly in Figure 12a. In fact, using SVET, it has already been reported that corrosion product deposition and partial deactivation occur heterogeneously for symmetrical assemblies [32,36], even with both galvanized sides also exposed in addition to the cut edge [42].

With longer durations, oxygen consumption increases again over the cut edge, resulting in the suppression of the cathodic current after 6 h as shown in Figure 11c and the blue line in Figure 12a. Then, over time, corrosion products formed on the galvanized layers and the iron surface apparently lose their protective character, probably due to the development of higher pH values that may lead to partial redissolution of the corrosion products forming oxy-hydroxide metal anions [37], leading to the appearance of additional vulnerable sites to sustain the cathodic reaction. Next, the oxygen diffusion profile, characterized by the lack of oxygen detected even far from the cathodically activated site, extends to a distance of 2000 µm from the cut edge after 24 h (cf. Figure 11d and green line in Figure 12a), revealing that the relatively high cathodic activity observed from the first hours occurred at a rate faster than oxygen diffusion can sustain. A similar effect has already been reported to a more limited extent when measuring embedded cut edge systems with larger (unrealistic) cathode-to-anode ratios, also leading to oxygen depletion in the neighbouring solution [38,39]. A greater effect is expected when the galvanized sides are not hindered by an insulating resin, which correlates well with the high oxygen consumption and generation of an extended diffusion layer observed here. This phenomenon would lead to a progression of the corrosive attack that is controlled by the diffusion of oxidizing species, in this case, the dissolved molecular oxygen.

The results obtained with the ZM90 coating are presented in Figure 13, revealing an apparently less localized oxygen consumption resulting in cathodic current values different from zero during the first hours of immersion (see Figure 13a). The appearance of white corrosion products prevented accurate measurement acquisition, and noisy results were obtained, as evidenced in the extracted lines of Figure 12b. The cut edge surface was poorly resolved throughout the experimental series, reflecting that oxygen consumption appears to occur not only over the iron-based cut edge, which is, in fact cathodically active, but is also observed in other areas where defects or delamination of the coating occurs within the first few hours. The current decay, observed to occur more progressively when compared with the ZM70 substrate, resulted in almost complete suppression of the oxygen content in solution after 6 h of immersion. This reflects an intense cathodic activation maintained for a longer time than when investigating the thinner coating.

Concerning the ZM120 sample, the recorded measurements seen in Figure 14 were severely affected by continuous deactivation of the probe due to the high reactivity of the sample, and the results could, therefore, not be taken into quantitative consideration. The thicker coating resulted in an even more extensive formation of visible white corrosion products, associated with the presence mainly of zinc oxy-hydroxides, and such interfering effect hindered the tip response. Therefore, reconditioning the tip by polishing and repositioning was necessary to continue scanning. Indeed, as the lines in Figure 14 were acquired at increasing *X* positions, the first lines obtained at the beginning of the exposure (eventually after each tip polishing step) provided measurable tip current, which next decays as a result of tip fouling. This excessive formation of corrosion products could be considered even detrimental to corrosion protection, as the surge of major heterogeneities and white rust at a faster rate may compromise the chemical and mechanical integrity of the material, alongside disadvantageous flexibility and adherence [57].

The local deposition of protective corrosion products must be associated primarily with the development of pH gradients and the presence of sufficient M^n+^ cations for the precipitation of metal hydroxides and/or carbonates [37]. To evaluate the effect of pH gradients developed during degradation at the exposed cut edge as vulnerable sites, antimony-based potentiometric microsensors were employed to monitor the changing the pH distributions [35]. The cathodic half-reaction is expected to consume hydrogen ions according to Equation (1), leading to an increase in pH over the cathodically activated areas. In contrast, the release of metal cations capable of undergoing hydrolysis can also compensate for this increase (or promote the hydrolysis reaction), thus leading to pH balance or acidification. Regarding Zn, this occurs by the following equilibrium reactions in an aqueous electrolyte:Zn → Zn^2+^ + 2e^−^(4)
(5)$${\mathrm{Zn}}^{2+}+2\;H_{2}O\;⇄\;\mathrm{Zn}{\left(\mathrm{OH}\right)}_{2}+2H^{+}$$

In addition, the presence of hydrogen carbonate in the SAR solution constitutes a source of carbonate anions, which can render the formation of hydoxycarbonates of the active metals. The associated equilibria are readily inferred by the presence of free CO_3_^2−^ in the solution [11,12], concurrently altered as a result of the local variations in pH:(6)$${{\mathrm{HCO}}_{3}}^{-}+{\mathrm{OH}}^{-}\;⇄\;{{\mathrm{CO}}_{3}}^{2-}+\;H_{2}O$$

As the main corrosion product, Zn(OH)_2_ is not expected to form at pH values below 5 according to the Pourbaix diagram [58], while Mg(OH)_2_ is only stable at a weakly alkaline pH (i.e., pH = 8). Therefore, the precipitation of these hydroxides in the SAR solution (pH = 4) is only possible under the establishment of a strongly alkaline pH due to the initiation of the cathodic half-reaction (1) on the nobler iron-based cut edge. Conversely, aluminium hydroxides and oxides, stable around neutral pH, should play a crucial role in pH balance and compensate for the expected alkalization over the cathode [37].

Figure 15 shows the pH distributions registered over the ZM70 cut edge in the SAR solution over time using the Sb-based electrode. The cathodic reaction (1) occurring on iron promotes alkalization above pH 10 at the early stages of the exposure (see Figure 15a), with fluctuations between 10.35 and 10.76 around the midpoint (*X* = 2000 µm) where the iron surface is located. This is also revealed when considering the evolution of the extracted linear scans at two different *Y* locations in Figure 16a. As seen in the map and the lines, the high pH value is roughly constant along the *Y* position at the centre of the cut edge. This pH is sufficient to promote the precipitation of zinc and magnesium hydroxides as they diffuse from the dissolving galvanized sides in solution to the iron surface at the cut edge, while the aluminium species would remain in solution. In addition, deeper valleys with a more acidic response are found, particularly at the start of the scan (positions *X* and *Y* equal to 0 µm), down to pH = 3.78, due to the hydrolysis of metal cations. This feature is rapidly compensated with the alkaline front from the iron surface, as revealed by the black solid line in Figure 16a, taken at *Y* = 100 µm. This observation and the fact that each map took almost 3 h to be completed, suggests that the acidification observed in the first recorded *X* lines reveals a temporal effect rather than a heterogeneous process.

From the beginning of the exposure (Figure 15a), alkalization readily occurs over the surface of the cut edge, and gradually leads to the diffusion of OH^−^ ions into the solution, resulting in neutralization of the acidic pH of the SAR test solution. Indeed, the line recorded for the same map at *Y* = 350 µm (dashed black line in Figure 16a) reveals not only an absence of acidification, but also the nearby environment is clearly attaining pH values much higher than that corresponding to the bulk solution (i.e., higher than pH 4), despite the tip not scanning over any reactive surface when placed at distant *X* positions in each of the lines. This is further supported by the continued tendency of the pH to increase at locations away from the cut edge after longer exposures, since the measured pH reaches an almost neutral value after 3 h at both sides of the cut edge, and this trend continues after 6 h of immersion (Figure 15b and Figure 15c, respectively). The lines in Figure 16 evidence pH range 7–8 far from the iron surface, regardless of the specific line, and significant diffusion of OH^−^ seems to occur towards negative values of the *X* axis after 3 h of immersion. However, alkalization over the iron surface still favours a pH range up to near 11, suggesting that the galvanic protection evidenced by the cathodic activation still remains active. This correlates well with the oxygen consumption observed when operating with the RC-SECM (cf. Figure 11), and also reveals cathodic activity (i.e., absence of oxygen) around the cut edge, this effect extending to longer distances from the edge as time elapses due to diffusion.

The pH micropotentiometric measurements obtained with the ZM90 system revealed similar features, as shown in Figure 17, but quantitative differences could be encountered. First, cathodic activation did not occur as early as with ZM70, evidenced by the fact that the alkalization over the middle portion of the scan only reached a pH near 10 at the end of the first measurement (Figure 17a). Indeed, the black lines in Figure 16b revealed pH values below 8 for the initial stages of the map acquisition. The cathodic activation remained in this range during subsequent scans (Figure 17b,c), revealing lower reactivity than for the ZM70 system with alkalization near pH 11 (see comparison of absolute pH values in Figure 16a,b). Local acidification near the galvanized sides is found with values similar to the solution pH in the first scan, with a minimum reading of 3.5 at specific spots, and clearly observed at the line extracted at *Y* = 100 µm. This reveals that hydrolysis of the metal ions dissolved from the galvanized sides occurs in highly localized regions and that the pH equilibrates rapidly in the solution. This observation was also acquired to a lesser extent for the ZM70 system in Figure 15a, and must be correlated with the observation of hydrogen bubbles during the SVET measurements, which would necessarily be accompanied by local acidification.

Heterogeneous but moderate diffusion of hydroxide anions is also observed at immersion times of 3 and 6 h, thereby increasing pH in regions where only bulk solution was expected, but not to the same neutral environment as previously observed for the ZM70 system (by comparing the maximum pH values from extracted lines in Figure 16a,b). In general, it is possible to assume that the greater formation of corrosion products results in a smoother evolution in the system and the protection of the cut edge when exposed in solution, although local activation of severe corrosion sites may result in the acidification of the iron-coating interface.

The analysis of the ZM120 was again affected by the significant deposition of white corrosion products, which prevented the reproducibility of the results. This must correlate with the increased coating thickness and the greater presence of Zn available for oxidation and subsequent precipitation, which is apparently accompanied by lower reactivity at the cut edge because it becomes protected by the precipitation of metal hydroxides. It was, though, possible to obtain information on the system when exposed to the SAR solution, as shown in Figure 18 and the extracted lines in Figure 16c.

The initial pH data recorded with the ZM120 system also reflects a dynamic evolution, as the pH at the beginning of the map was found to be less stable than upon scan completion. Indeed, a pH as low as 2.77 was found in both valleys flanking the cut edge, revealing an intense release of metal cations prone to hydrolysis and the formation of corrosion products, accompanied by the observation of the hydrogen gas evolution. These valleys were rather asymmetric, as shown in Figure 16c (black solid line taken at *Y* = 200 µm), and the effect was compensated after 6 h of immersion, resulting in a smooth evolution of the pH profile reflecting general cathodic activation of the cut edge. This is also associated with less pronounced alkalization over the iron edge as compared with the less dense coating systems, which may actually result in delayed pH compensation, thus explaining the local acidic value observed here. Indeed, once the scan was completed, the diffusion of the OH^−^ generated at the edge was sufficient to increase the pH recorded away from the cut edge, and the last lines exhibited pH values higher than that of the bulk solution.

## 4. Conclusions

The microstructural domains of advanced hot-dip ZnAlMg galvanized coatings on mild steel were investigated and correlated with their sacrificial and passivation protection effects following cut edge exposure under realistic exposure conditions, that is, concomitantly exposing the lateral galvanized sides to the aggressive electrolyte. Three coating thicknesses were investigated and compared, namely ZM70, ZM90 and ZM120, with coating densities of 70 g m^−2^, 90 g m^−2^ and 120 g m^−2^, respectively. The following conclusions can be extracted:The thinnest ZM70 system revealed accumulation of Al and Mg as binary and ternary eutectics, as well as oxides, whereas Zn was mostly present in its metallic formed. Conversely, denser ZM90 and ZM120 coatings revealed a higher presence of zinc oxides and hydroxides, according to SEM-EDS and GDOES, associated with the appearance of white corrosion products.Mg and Al rather accumulated as eutectic phases, but also at the iron-coating interface, in particular for denser coating thicknesses. This promoted highly localized anodic behaviour from these sites, as revealed by SVET and pH micropotentiometry, accompanied with hydrogen evolution, evidencing severe localized degradation.The cathodic activity over the cut edge was found to progress heterogeneously, with local activation at spots where no deposition of white corrosion products occurred. The attained alkaline pH above the cathode would favour the precipitation of zinc and magnesium hydroxides and carbonates, which was particularly intense in the ZM90 and ZM120 systems. The oxygen consumption and alkalization resulted in the oxygen exhaustion and neutralization of the nearby electrolyte in simulated acid rain, thus maintaining favourable conditions for surface passivation in all systems.

Overall, the economically favourable development of the thinnest ZM70 system, combined with lower consumption of reactive metal in the molten bath, provides sufficient short-term protection and reduced localized acidification at the iron-coating interface that is associated with severe intense corrosion processes. This renders promising results for this lesser dense coating, as benefits are also envisaged regarding adherence, uniformity and flexibility against thermomechanical stress. The surface roughness of the ZM70 was actually comparable to that of the thicker ZM120 system, which concurrently led to the active formation of white snowflakes, which affect the measurements and the environment due to intense reactivity. Certainly, the corrosion products formed exhibit a greater protective behaviour for the ZM90 and ZM120 systems; however, the protection and reactivity decay of the ZM70 coating is expected to repassivate a given cut edge exposed to aggressive conditions by maintaining a sufficiently alkaline environment for the metal hydroxides and hydroxycarbonates to precipitate. An initially higher dissolution rate for the ZM70 may indeed prevent the formation of visible and mechanically undesired white rust, while still providing cathodic protection and more efficiently preventing severe corrosion locally at the iron-coating interface. Therefore, a thinner galvanized coating may offer superior long-term corrosion resistance compared to thicker ones.

## Figures and Tables

**Figure 1 materials-17-01679-f001:**
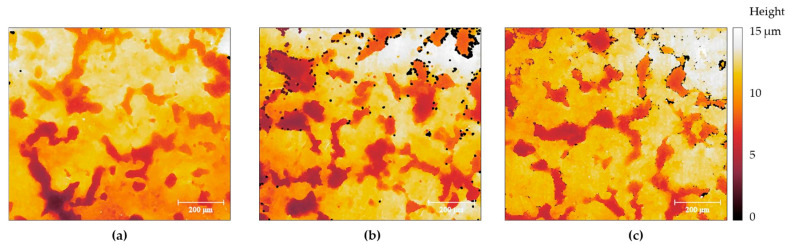
Morphological images obtained with an interferometry-based profilometer for (**a**) ZM70, (**b**) ZM90 and, (**c**) ZM120 surfaces. Images were levelled and denoised using the Gwyddion closing algorithm down to 5 px. size. The image size is 960 × 840 µm^2^.

**Figure 2 materials-17-01679-f002:**
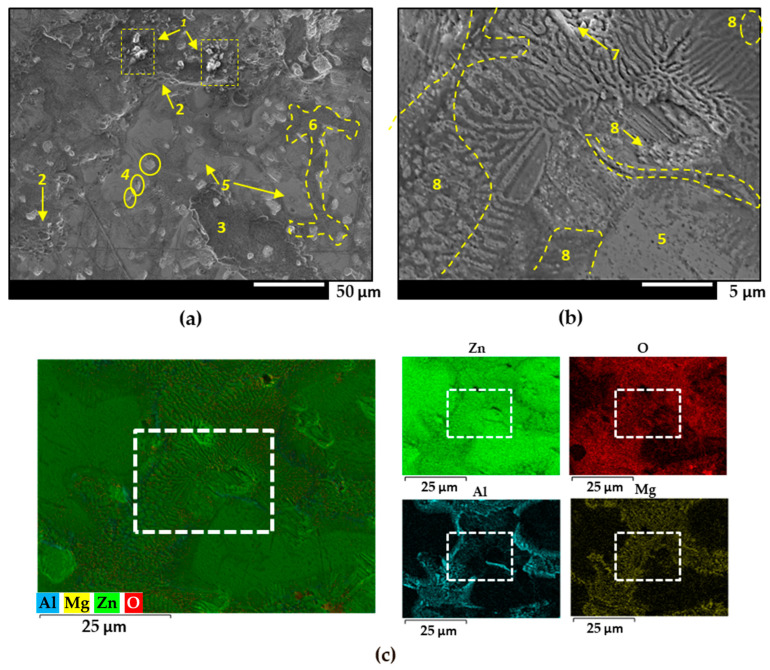
(**a**,**b**) FEG-SEM images of the ZM70 coating at different magnifications. (**c**) Semiquantitative EDS analysis and individual elemental maps of the same area. The white squares in maps of (**c**) corresponds to the area seen in (**b**). For clarifications to the yellow labels in (**a**,**b**), the reader is referred to the text.

**Figure 3 materials-17-01679-f003:**
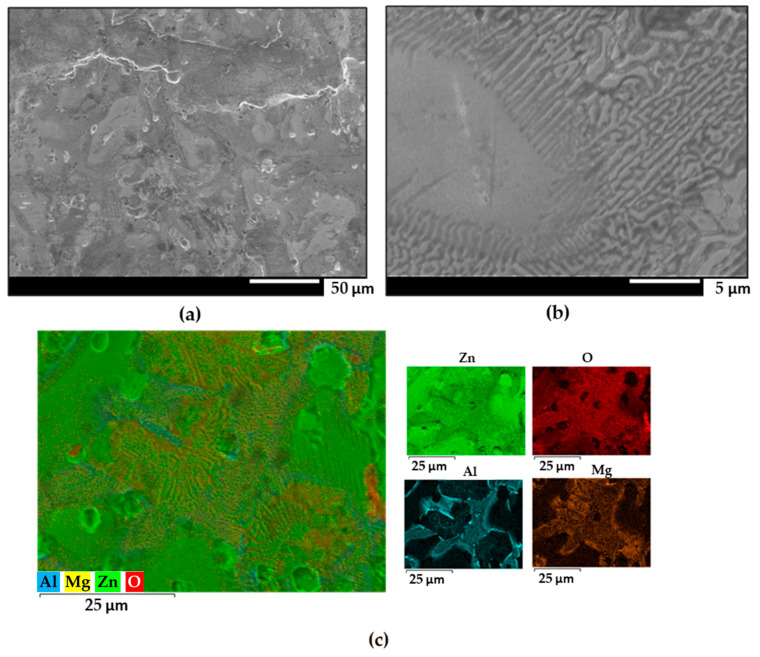
(**a**,**b**) FEG-SEM images of the ZM90 coating at different magnifications. (**c**) Semiquantitative EDS analysis and individual elemental maps of the same area.

**Figure 4 materials-17-01679-f004:**
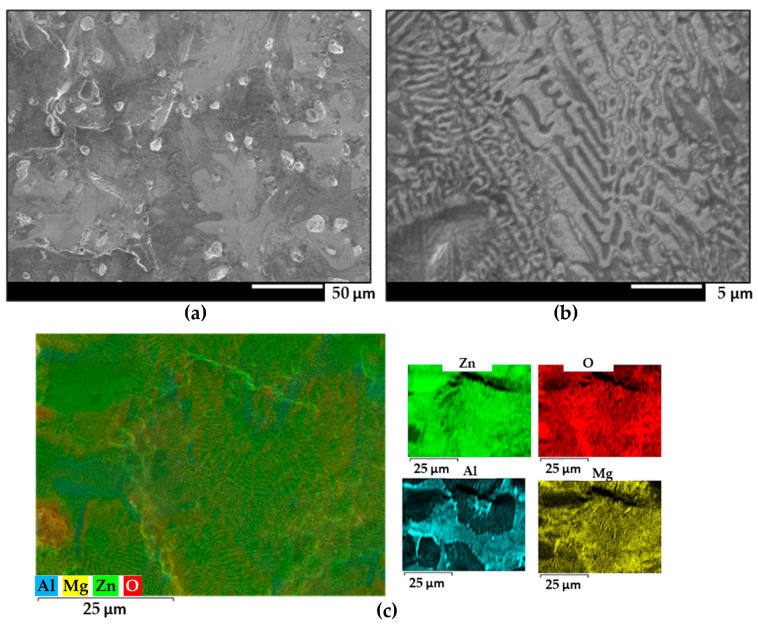
(**a**,**b**) FEG-SEM images of the ZM120 coating at different magnifications. (**c**) Semiquantitative EDS analysis and individual elemental maps of the same area.

**Figure 5 materials-17-01679-f005:**
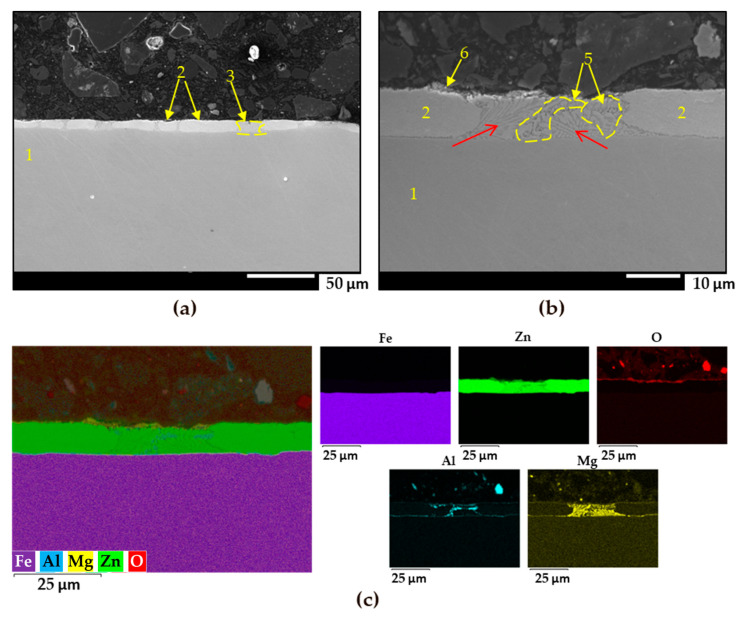
(**a**,**b**) FEG-SEM images of the cross-section of the ZM70 coating on steel at different magnifications. (**c**) Semiquantitative EDS analysis and individual elemental maps of the same area. For clarifications to the yellow labels and red arrows in (**a**,**b**), the reader is referred to the text.

**Figure 6 materials-17-01679-f006:**
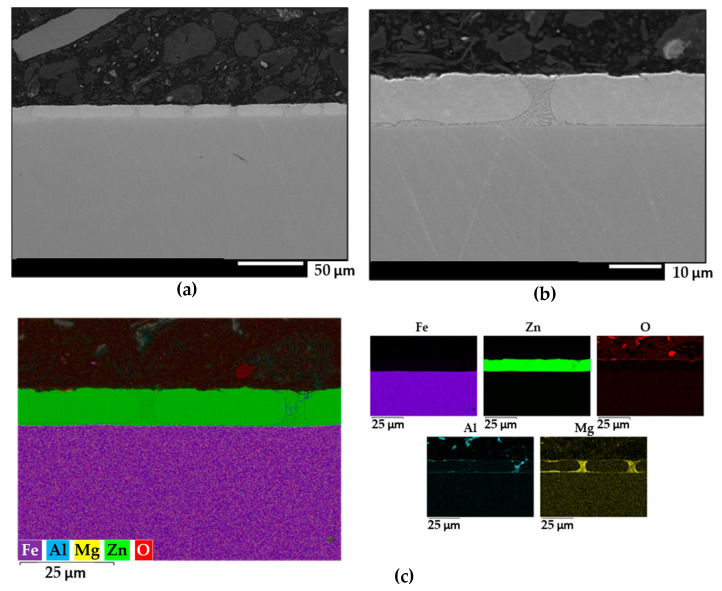
(**a**,**b**) FEG-SEM images of the cross-section of the ZM90 coating on steel at different magnifications. (**c**) Semiquantitative EDS analysis and individual elemental maps of the same area.

**Figure 7 materials-17-01679-f007:**
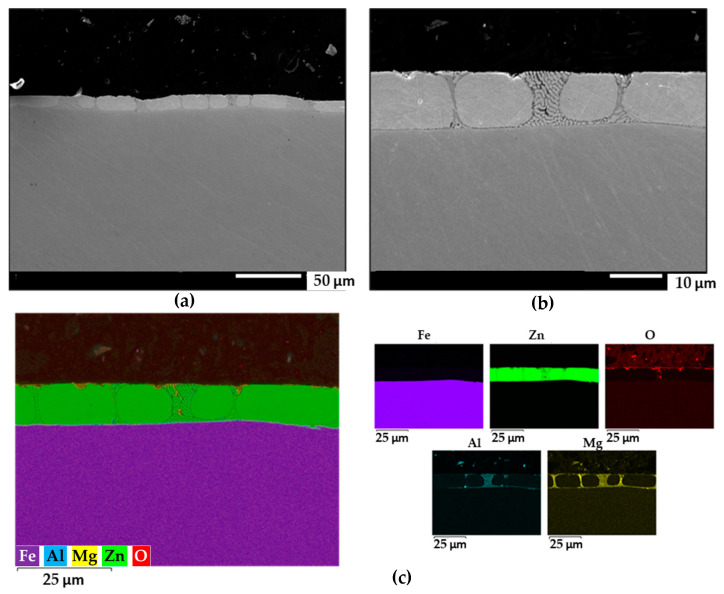
(**a**,**b**) FEG-SEM images of the cross-section of the ZM120 coating on steel at different magnifications. (**c**) Semiquantitative EDS analysis and individual elemental maps of the same area.

**Figure 8 materials-17-01679-f008:**
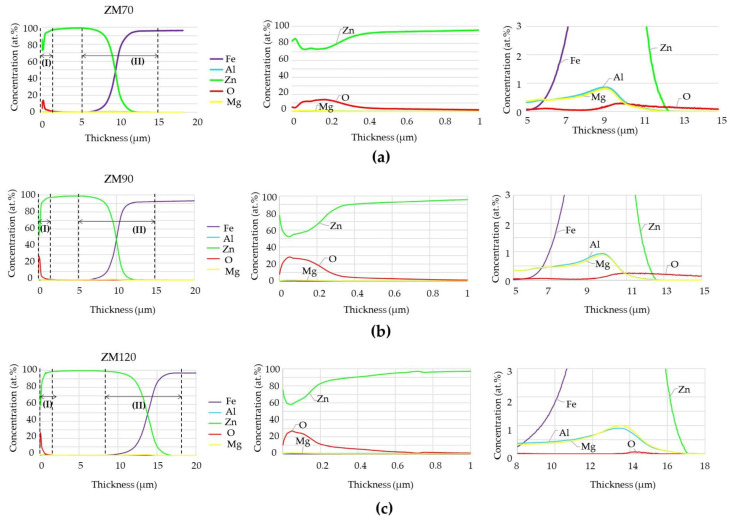
GDOES profiles acquired for (**a**) ZM70, (**b**) ZM90 and (**c**) ZM120 coatings. The colour codes refer to the content of (**–**) Fe, (**–**) Zn, (**–**) O, (**–**) Al and (**–**) Mg.

**Figure 9 materials-17-01679-f009:**
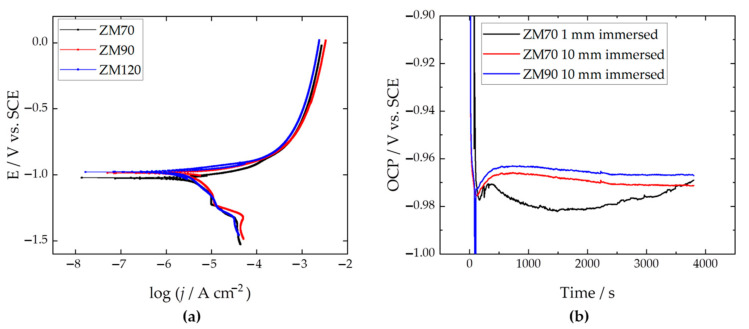
(**a**) Dynamic polarization curves obtained with the galvanized surface exposed to the SAR test solution after 3000 s of immersion at a sweep rate of 1 mV s^−1^. (**b**) OCP recorded during the first hour by immersing the edge of ZM70 and ZM90 samples to approximate immersion depths of 1 or 10 mm, as indicated in the legend.

**Figure 10 materials-17-01679-f010:**
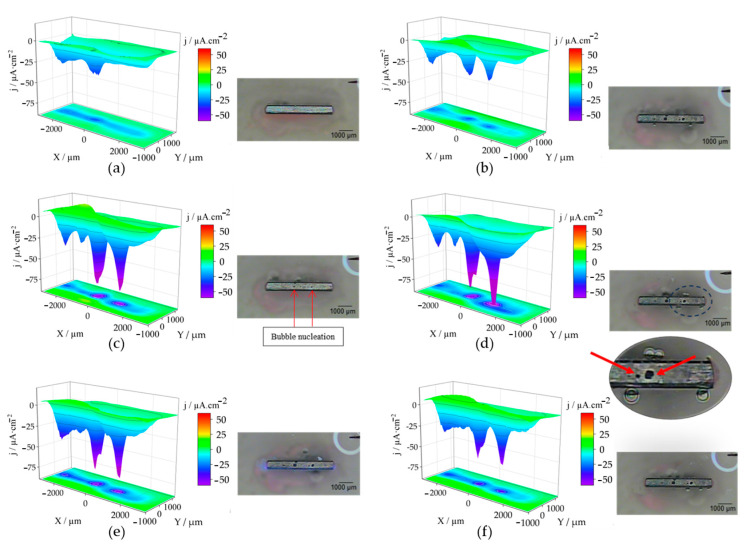
SVET images of a cut edge of ZM70 immersed in SAR solution for different times: (**a**) 0 h, (**b**) 8, (**c**) 16, (**d**) 24, (**e**) 32 and (**f**) 36 h. Alongside, the optical micrographs taken after completing SVET map acquisition. Red arrows point at sites of interest: hydrogen evolution in (**c**), active spot uncovered with corrosion products in (**d**). Each scan lasted approximately 15 min.

**Figure 11 materials-17-01679-f011:**
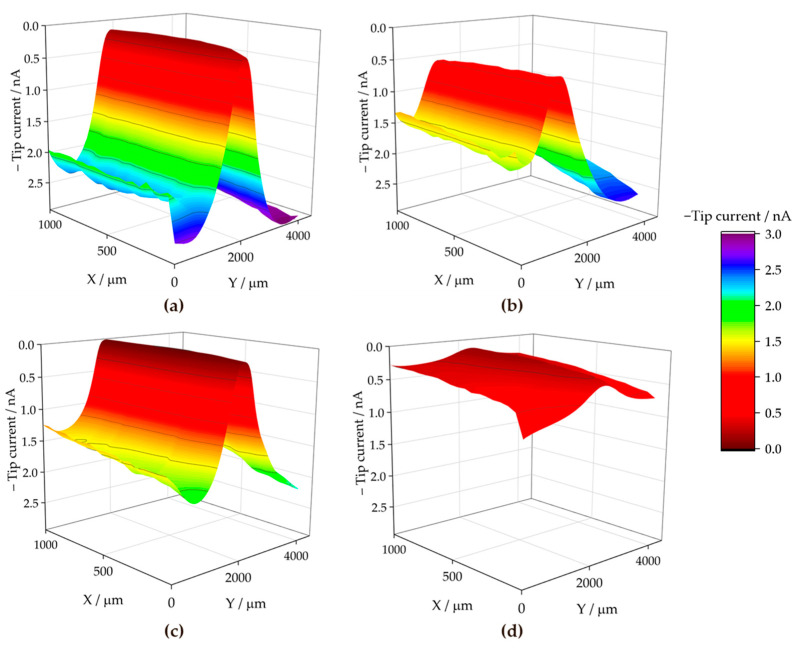
RC-SECM maps recorded over the cross-section of a ZM70 sample immersed in SAR solution for different immersion times: (**a**) 0, (**b**) 3, (**c**) 6 and (**d**) 24 h. Tip potential: −0.75 V vs. SCE; tip-substrate distance: 50 µm; scan rate: 20 µm s^−1^. Each scan lasted approximately 160 min.

**Figure 12 materials-17-01679-f012:**
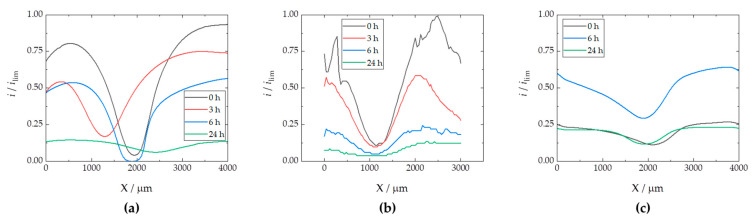
Linear scans extracted from the middle lines of the RC-SECM maps obtained with the (**a**) ZM70, (**b**) ZM90 and (**c**) ZM120 samples immersed in SAR solution. *i*_lim_: limiting current for oxygen reduction at the tip in bulk solution: (**a**) −2.94, (**b**) −0.82 and (**c**) −1.12 nA.

**Figure 13 materials-17-01679-f013:**
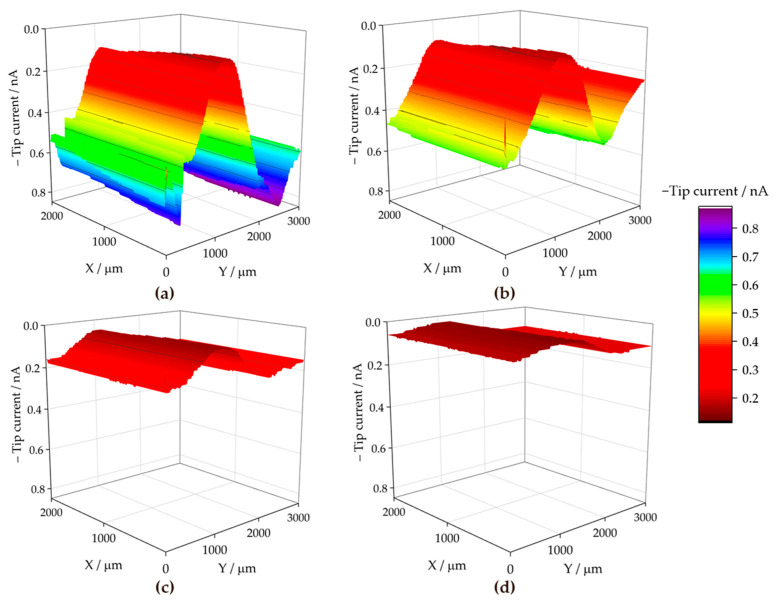
RC-SECM maps recorded over the cross-section of a ZM90 sample immersed in SAR solution for different immersion times: (**a**) 0, (**b**) 3, (**c**) 6 and (**d**) 24 h. Tip potential: −0.75 V vs. SCE; tip-substrate distance: 50 µm; scan rate 20 µm s^−1^. Each scan lasted approximately 160 min.

**Figure 14 materials-17-01679-f014:**
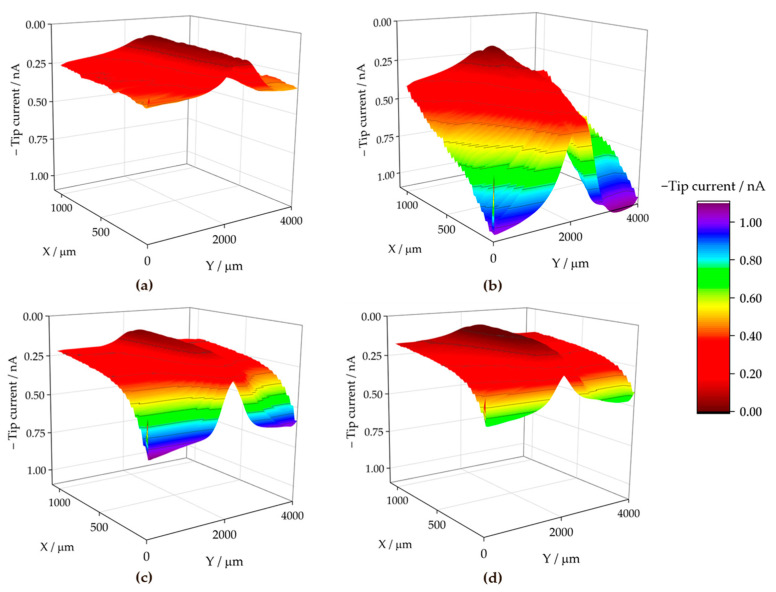
RC-SECM maps recorded over the cross-section of a ZM120 sample immersed in SAR solution for different immersion times: (**a**) 0, (**b**) 6, (**c**) 24 and (**d**) 48 h. Tip potential: −0.75 V vs. SCE; tip-substrate distance: 50 µm; scan rate 20 µm s^−1^. Each scan lasted approximately 160 min.

**Figure 15 materials-17-01679-f015:**
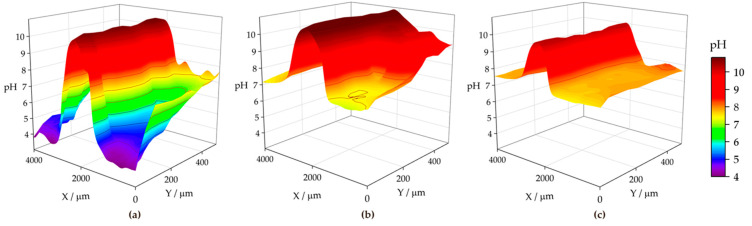
pH maps recorded over the cross-section of a ZM70 sample immersed in SAR solution for different immersion times: (**a**) 0, (**b**) 3, and (**c**) 6 h. Tip-substrate distance: 50 µm; scan rate: 20 µm s^−1^. Each scan lasted approximately 160 min.

**Figure 16 materials-17-01679-f016:**
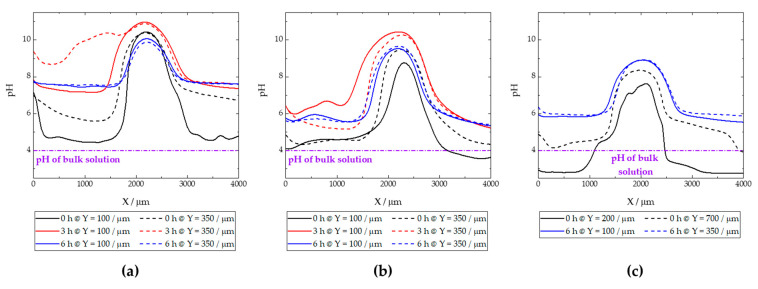
Linear scans extracted from two selected lines of the micropotentiometric pH maps obtained with the (**a**) ZM70, (**b**) ZM90 and (**c**) ZM120 samples immersed in SAR solution. The *Y* position for the line extraction is indicated in the legend.

**Figure 17 materials-17-01679-f017:**
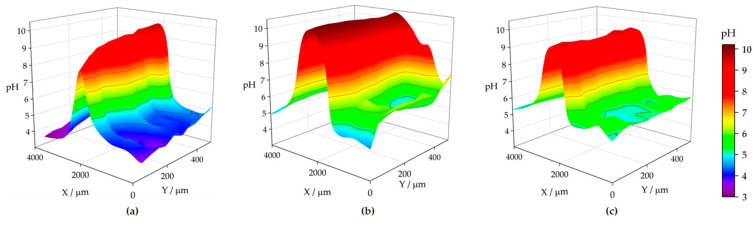
pH maps recorded over the cross-section of a ZM90 sample immersed in SAR solution for different immersion times: (**a**) 0, (**b**) 3, and (**c**) 6 h, Tip-substrate distance: 50 µm; scan rate: 20 µm s^−1^. Each scan lasted approximately 160 min. Red colour is used for alkaline pH, blue colour is used for acidic pH.

**Figure 18 materials-17-01679-f018:**
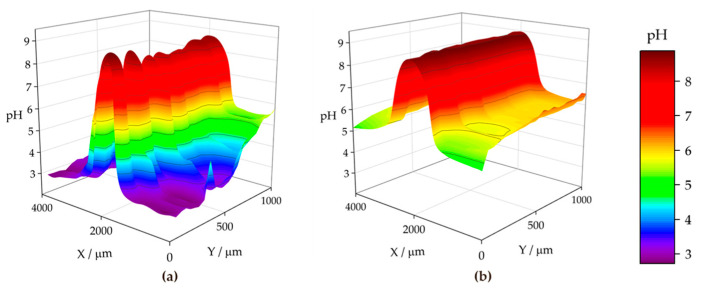
pH maps recorded over the cross-section of a ZM120 sample immersed in SAR solution for different immersion times: (**a**) 0, and (**b**) 6 h. Tip-substrate distance: 50 µm; scan rate: 20 µm s^−1^. Each scan lasted approximately 400 min.

**Table 1 materials-17-01679-t001:** Composition of the DX57D steel material in % wt.

C	Mn	P	S	Si	Ti	Fe
≤0.120	≤0.60	≤0.100	≤0.045	≤0.50	≤0.300	Balance

**Table 2 materials-17-01679-t002:** Corrosion parameters extracted by Tafel analysis of the polarization curves in Figure 9a.

	*E*_corr_/V	*j*_corr_/µA cm^−2^	−*β*_cat_/mV dec^−1^	*β*_an_/mV dec^−1^
ZM70	−0.976	3.17	244	55.3
ZM90	−0.935	3.37	277	56.4
ZM120	−0.927	2.70	277	53.1

## Data Availability

The raw/processed data required to reproduce these findings cannot be shared at this time as the data also form part of an ongoing study. They will be available on request.

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
