# Peer review of "Contributions to a More Realistic Characterization of Corrosion Processes on Cut Edges of Coated Metals Using Scanning Microelectrochemical Techniques, Illustrated by the Case of ZnAlMg-Galvanized Steel with Different Coating Densities"

_materials, 2024, doi:10.3390/ma17071679_

Round 1
Reviewer 1 Report
Comments and Suggestions for Authors
1. In the abstract : the description of the work content is not clear, and the description of the experimental results should be reflected by quantitative analysis.
2. It is recommended to display the chemical composition of ZnAlMg galvanized steels in the form of a table.
3. As shown in figure 1, black spots appeared on the surface of ZM70, ZM90 and ZM120, and black spots were the most distributed in ZM90. It is necessary to explain what they are and analyze the reasons for their emergence.
4. In 282 lines. ' Interesting observations can be made by cross-referencing the image in Figure 2b with the aluminum and magnesium contents revealed by EDS in Figure 2c ' is mentioned. However, the contents of Al and Mg are not shown, and Figure 2b and Figure 2c are not pictures in the same position. The paragraph needs to be modified.
5. In lines 307-317, ' Figures S1 and S2 ' is mentioned. However, they are not found in the article.
6. Adjust the ruler in the picture ( Figure 2, Figure 3 ) to ensure that the adjusted format is the same
7. In line 329-336, the location of the label description is uncertain.
8. As shown in Figure 6, the ion current density decreased with time at 24 h. The reason for this phenomenon has not been explained.
9. For the sample ZM120, the following pictures need to be supplemented : the pH map recorded on the cross section after soaking in SAR solution for 3h and RC-SECM figure.
Author Response
Color code:
- Black: reviewer’s query
- Green: answer to the reviewer –
We thank the reviewer for the time dedication, and we consider that the manuscript has been improved following not only queries but also the comments in the global evaluation table. The introduction and referencing have been revised according to the constructive critics from the table and specific queries pointed by the reviewers. References have been updated and relevant literature on the optimization of the coatings composition for the automotive industry included. Other modifications have been conducted following the reviewers comments, such as the clarification of the materials composition and minor helping comments regarding the methodology. We also expect that the display of linear-scans and the inclusion of the figures previously presented in the supporting information have improved the quality of the manuscript.
- In the abstract: the description of the work content is not clear, and the description of the experimental results should be reflected by quantitative analysis.
The abstract has been rewritten so that the scope of the employed techniques is more clearly stated, and relevant quantitative data are now provided in terms of time required for oxygen complete depletion and pH neutralization, as well as approximate pH values locally attained.
- It is recommended to display the chemical composition of ZnAlMg galvanized steels in the form of a table.
As suggested, the composition has been included as Table 1.
- As shown in figure 1, black spots appeared on the surface of ZM70, ZM90 and ZM120, and black spots were the most distributed in ZM90. It is necessary to explain what they are and analyze the reasons for their emergence.
The black spots stem from the de-noise algorithm, that was referred in the Figure caption. This information is now further clarified in the text.
- In 282 lines. ' Interesting observations can be made by cross-referencing the image in Figure 2b with the aluminum and magnesium contents revealed by EDS in Figure 2c ' is mentioned. However, the contents of Al and Mg are not shown, and Figure 2b and Figure 2c are not pictures in the same position. The paragraph needs to be modified.
We thank the reviewer for noticing this confusing statement, relevant for the discussion. The area imaged in Figure 2b is indeed a magnification of a region of interest inside that mapped in Figure 2c. A new version of Figure 2 is now provided, where Figure 2c has been labeled with white squares in order to reflect the location of the zoomed area for Figure 2b, and the text has been modified accordingly.
- In lines 307-317, ' Figures S1 and S2 ' is mentioned. However, they are not found in the article.
Figures S1 and S2 were found in the supplementary materials file. In order to facilitate discussion, and agreeing that they are relevant in the work, we have included the figures in the main manuscript as Figures 3 and 4.
- Adjust the ruler in the picture ( Figure 2, Figure 3 ) to ensure that the adjusted format is the same
The rulers have been revised and their position corrected.
- In line 329-336, the location of the label description is uncertain.
We thank the reviewer for noticing this confusing description. Labels have been revised and corrected, so that now one can ascribe the same feature in both images (new figures 5a and 5b) using the same label.
- As shown in Figure 6, the ion current density decreased with time at 24 h. The reason for this phenomenon has not been explained.
The reason for the progressive decay in SVET response relates with the formation of protective corrosion products, responsible for the surface passivation. Indeed, the formation of white corrosion products (typically zinc hydroxides, oxides and carbonates) is next mentioned. This correlation is more clearly stated in the new version of the manuscript.
- For the sample ZM120, the following pictures need to be supplemented : the pH map recorded on the cross section after soaking in SAR solution for 3h and RC-SECM figure.
As mentioned in the manuscript, SECM mapping of the ZM120 sample was, regardless of the operation mode, challenging due to the intense formation of corrosion products that hindered the tip response while conducting the surface analysis. It is claimed that The RC-SECM “was affected by continuous deactivation of the probe due to the high reactivity of the sample”, hence maps cannot be entirely taken into consideration and were originally included in the supplementary material file. The maps corresponding to the clearer experimental series are now included for reference as Figure 14, and the limitations observed during scan acquisition are commented in the text.
As micropotentiometric measurements for pH imaging in ZM120 regards, it is referred that it “was again affected by the significant deposition of white corrosion products, which prevented reproducibility of the results”. Such loss of reproducibility prevented the acquisition of measurements exactly under the same conditions, and the experimental series providing the clearest results is now shown as Figure 18. This series conveyed longer maps (4000 x 1000 µm2, vs. 4000 x 5000 µm2 for Figures 15 and 17), thus requiring longer acquisition time and rendering impossible to obtain a scan measurement after 3 hours immersion. The approximate acquisition times for each map is stated in the corresponding figure captions.
Reviewer 2 Report
Comments and Suggestions for Authors
The reviewed paper titled “Contributions to a more realistic characterization of corrosion processes on cut edges of coated metals using scanning micro electrochemical techniques, illustrated by the case of ZnAlMg-galvanized steel with different coating densities” is an interesting work associated with corrosion resistance of steel after hot-dip galvanizing. The topics discussed in the work are very current. The results are of great importance not only from a scientific point of view but also from a practical point of view (application in industry).
The authors did not avoid minor errors and inaccuracies. Notes for reflection and minor correction of the work:
Keywords - word “mild”- remove
1. Introduction
The literature review lacks information on the chemical compositions of coatings used on hot-dip galvanized steel products. the authors indicate ZnAlMg coatings, but there are also Zn and ZnAl coatings. The percentage of elements in zinc coatings, e.g. Al, ranges from 2 to even 10%, and magnesium from 0.5 to over 5%.
2. Materials and Methods
- according to what standard was the corrosion test carried out?
- in my opinion, the value of the work would also be increased by testing the corrosion resistance of samples with different coating thicknesses in a salt chamber.
- unfortunately the authors did not compare the test results of ZnAlMg coatings with the standard Zn coating. If possible, it would be good to show such a comparison at work
3. Results and discussion
- the descriptions in Figure 4 are illegible
- chapter 3 is missing simple linear figures illustrating the influence of the thickness of the galvanized coating on the change of the x, y, x parameter... e.g. on the pH parameter at a constant value of time
4. Conclusions
Chapter 4 is not a conclusion, but a rather extensive summary. I kindly ask you to redraft this part of the work and formulate concise conclusions
Author Response
Color code:
- Black: reviewer’s query
- Green: answer to the reviewer
The reviewed paper titled “Contributions to a more realistic characterization of corrosion processes on cut edges of coated metals using scanning micro electrochemical techniques, illustrated by the case of ZnAlMg-galvanized steel with different coating densities” is an interesting work associated with corrosion resistance of steel after hot-dip galvanizing. The topics discussed in the work are very current. The results are of great importance not only from a scientific point of view but also from a practical point of view (application in industry).
We thank the reviewer for the positive comments and recognition regarding the relevance of the work. Following also the global evaluation in the table and comments from this reviewer, the conclusions are now more concisely presented as bulleted points, aiming to provide proper interpretation of the systems considering the convergence of all the experimental observations.
The authors did not avoid minor errors and inaccuracies. Notes for reflection and minor correction of the work:
Keywords - word “mild”- remove
Done, the word “steel” is now a keyword itself.
- Introduction
The literature review lacks information on the chemical compositions of coatings used on hot-dip galvanized steel products. the authors indicate ZnAlMg coatings, but there are also Zn and ZnAl coatings. The percentage of elements in zinc coatings, e.g. Al, ranges from 2 to even 10%, and magnesium from 0.5 to over 5%.
We thank the reviewer for this suggestion. We have included comments on the optimization of the coatings composition in the second page as part of the introduction, along with corresponding references. According to the scientific literature, the composition of the coatings as optimized for the automotive industry is very close to the herein tested commercial materials, being ZAM coatings typically based on Zn-3Al-3Mg or similar, with far superior anticorrosion performance than traditional HDG, Galfan or Galvalume.
- Materials and Methods
- according to what standard was the corrosion test carried out?
Electrochemical methods were conducted according to current knowledge in electrochemical phenomena. Some methods are neither standard nor widely applied, as it occurs with the recording of the Open Circuit Potential with the sample partially or totally immersed in solution. However, the observations revealed interesting trends that deserved discussion and interpretation. The fact that this strategy was not conventional is now stated in the text.
As the dynamic polarization and the microelectrochemical methods regard, although no standard regulation can be referred, the methods respond to widely applied electrochemical routines for the assessment of the anti-corrosion materials performance. Dynamic polarization methods reveal the inherent electrochemical phenomena behind a given galvanic corrosion process. As further advanced techniques gathering local information, SECM and SVET are worldwide recognized as powerful tools for the visualization of local corrosion phenomena, and this is fully depicted along the cited references and reviews.
- in my opinion, the value of the work would also be increased by testing the corrosion resistance of samples with different coating thicknesses in a salt chamber.
We agree that this strategy would enhance the scientific value of the work in terms of accelerated test for the evaluation of the anticorrosion performance under atmospheric exposure. However, being a commercially available material with proven efficiency against typical exposure conditions, our scope in this manuscript was to address possible mechanistic insights that would describe and predict the materials protection and self-repair under cut edge exposure configuration, in particular with respect to local oxygen consumption and pH gradients developed over a realistically exposed cut edge. The correlation with these phenomena with the microstructure domains seems crucial for the understanding of the coating performance. The evaluation of the global response in macroscopic terms, as it would be provided by salt-spray test, does not convey with the main scope of the manuscript.
- unfortunately the authors did not compare the test results of ZnAlMg coatings with the standard Zn coating. If possible, it would be good to show such a comparison at work
Again, being the material a commercially available system, it is certainly advanced with respect to conventional Zn coatings. Recent advances have demonstrated that ZnAlMg coatings provide greater protection efficiency through the formation of new phases and rapid passivation, and this is more clearly illustrated in the new comments included in the introduction regarding the optimization of the coating composition. Prospect coatings of industrial interest are expected to operate with optimized coatings compositions close to those herein considered, namely Zn-xAl-yMg (x and y lay around 2.5-3.5 according to references 15 to 21 in the new version of the manuscript). However, rather than comparing the performance with respect to other coatings, this work aimed at acquiring mechanistic insights on the galvanic interaction in these materials with complex microstructure, and relate the observations and expectations with the coating densities.
- Results and discussion
- the descriptions in Figure 4 are illegible
The figure has been edited accordingly, and we expect that it is now appropriate for reading and interpretation.
- chapter 3 is missing simple linear figures illustrating the influence of the thickness of the galvanized coating on the change of the x, y, x parameter... e.g. on the pH parameter at a constant value of time
Lines have been extracted following the reviewer recommendation, and inter-comparison of the systems seems now easier following Figures 12 and 16.
- Conclusions
Chapter 4 is not a conclusion, but a rather extensive summary. I kindly ask you to redraft this part of the work and formulate concise conclusions
We agree with the reviewer’s suggestion and we have redrafted the conclusion in the form of bulleted points to provide relevant observations relating the microstructural characterization and the electrochemical behavior, with particular emphasis in the new findings and observations. Emphasis is done in the new observations that were not expected from previous reports due to the different configuration of the cut edge exposure. The last paragraph has barely been modified as we understand that it provides conclusions based on the observations that advance in the optimization of coatings systems with greater protection efficiency even when cut edges are exposed.